# GAUGE EQUIVARIANT SPHERICAL CNNS

## ABSTRACT

Spherical CNNs are convolutional neural networks that can process signals on the sphere, such as global climate and weather patterns or omnidirectional images. Over the last few years, a number of spherical convolution methods have been proposed, based on generalized spherical FFTs, graph convolutions, and other ideas. However, none of these methods is simultaneously equivariant to 3D rotations, able to detect anisotropic patterns, computationally efficient, agnostic to the type of sample grid used, and able to deal with signals defined on only a part of the sphere. To address these limitations, we introduce the Gauge Equivariant Spherical CNN. Our method is based on the recently proposed theory of Gauge Equivariant CNNs, which is in principle applicable to signals on any manifold, and which can be computed on any set of local charts covering all of the manifold or only part of it. In this paper we show how this method can be implemented efficiently for the sphere, and show that the resulting method is fast, numerically accurate, and achieves good results on the widely used benchmark problems of climate pattern segmentation and omnidirectional semantic segmentation.

## 1 INTRODUCTION

In many disciplines of science and engineering, spherical signals emerge naturally. In the earth and climate sciences, globally distributed sensor arrays collect measurements like temperature, pressure, wind directions, and many other variables. Cosmologists are interested in identifying physical model parameters from real and simulated cosmic microwave background measurements sampled on spherical sky maps. In robotics, especially in applications like SLAM and visual odometry, omnidirectional and fish-eye cameras are widely used. Thus, it is clear that efficient CNNs that can directly operate on spherical signals are necessary.

When designing Spherical CNNs for these applications, there are a number of practical considerations that need to be taken into account. Firstly, whereas planar images are always sampled on a square grid, there is a wide variety of spherical grids, and the choice of sampling grid is usually dictated by hardware or task-dependent requirements. Secondly, several applications require the analysis of very high resolution spherical maps. Most existing algorithms cannot handle this scale. Thirdly, in some cases data is only available on a part of the sphere. Fourthly, whereas several graph-based and spectral methods can only learn isotropic blob-like filters, some problems require the detection of anisotropic patterns. Finally, in almost all cases, equivariance to rotations is a sought-after property since results should never depend on how we orient the sphere.

The method we propose in this paper is based on the recently proposed theory of Gauge CNNs (Cohen et al., 2019). This framework has several appealing properties: *(i)* it provides a principled way to achieve spatial and rotational weight-sharing on the sphere, *(ii)* it naturally allows for processing geometric features such as scalar, vector and tensor fields in an anisotropic and equivariant manner, and *(iii)* it makes it possible in principle to process a signal on overlapping local charts, in case data is not available for the whole sphere or when the whole signal does not fit in memory. To the best of our knowledge, we present the first implementation of a Gauge CNNs on a manifold with non-trivial curvature. We describe an efficient implementation which is, in principle, applicable to any grid on the sphere. Experiments on spherical segmentation of climate patterns and omnidirectional images demonstrate the effectiveness of our method. We show that our method is scalable and fast while being as numerically accurate as the best existing Spherical CNNs.

## 2    RELATED WORK

**Conventional CNNs for Spherical Images.** A line of prior work (Su & Grauman, 2017; Coors et al., 2018) considers using planar CNNs on intrinsically spherical tasks such as omnidirectional image segmentation. These approaches represent the spherical data on $\mathbb{R}^2$ via equirectangular projection which introduces position-dependent distortion rates due to curvature. In order to counteract the varying distortion rates, Su & Grauman (2017) uses kernels with increasing support towards the poles thus limiting parameter-sharing to only longitudes. Contrastingly, Coors et al. (2018) use fixed-sized kernels sampled on the tangent plane of spherical signal for better parameter-sharing and distortion invariance. Both methods assume a preferred orientation and are not SO(3) equivariant.

**Spherical CNNs.** Another line of work operates directly on $S^2$ alleviating the issues related to distortions caused by projection. Boomsma & Frellsen (2017) proposes spherical convolutions for molecular modeling in both volumetric and mesh representations. Perraudin et al. (2018) proposes a graph-based spherical convolution for cosmological model classification. A subset of this line (Kondor et al., 2018; Cohen et al., 2018; Esteves et al., 2018) extends group equivariance (Cohen & Welling (2016)) over SO(3). Esteves et al. (2018) uses spherical convolutions computed by spherical FFTs. Similarly, Cohen et al. (2018) proposes a spherical correlation operation based on the SO(3)-FFT. As a result both methods have inbuilt rotation equivariance property. However, the Generalized FFT algorithms used only work on inhomogeneous grids that over-sample the poles, and in practice the FFT-based methods are slow.

Recently, a number of spherical CNNs that parametrize the sphere with icosahedral tiling have been proposed (Jiang et al., 2019; Zhang et al., 2019; Liu et al., 2019; Cohen et al., 2019). These methods are tailored for the icosphere grid since this grid is quite regular and its charts can be easily mapped on $\mathbb{R}^2$ allowing highly optimized convolution routines to be used. Out of these methods, Gauge Equivariant Icosahedral CNN of Cohen et al. (2019) is the most similar to our method. However, their model operates on the icosahedron, which is only an approximation of the sphere, while ours operates on the sphere. Even though their method is fast and accurate, it is only equivariant to discrete icosahedral symmetries, while our method is fully SO(3)-equivariant. In comparison to all methods in this category, our method is agnostic to sampling grid used and better suited for larger scale problems.

**Geometric Deep Learning.** There have been a number of attempts to generalize convolution operator to manifolds in the geometric deep learning field (Bronstein et al., 2017). The key issue with manifold convolution is the lack of globally consistent reference frames attached to points over a manifold. Therefore, unlike shifting a filter over a flat image grid with a clear sense of up/down and left/right, it is not clear how to place the convolution kernel. To overcome this issue, Bruna et al. (2014); Boscaini et al. (2015) have used isotropic filters at the expense of kernel expressivity. Masci et al. (2015) have applied filters in fixed number of orientations and accumulated the responses via max pooling, losing the orientation information. Recently proposed Gauge Equivariant CNNs (Cohen et al., 2019) have shown that it is possible to have both expressive kernels and orientation information by allowing the network response change equivariantly with respect to arbitrarily chosen local reference frames (i.e. gauges).

## 3    CONTINUOUS THEORY OF GAUGE CNNS

In this section we will review the mathematical theory of gauge CNNs on general manifolds, as presented in Cohen et al. (2019). Specifically, we will define a mathematical model of feature spaces as fields, and show how one can define a convolution-like operation that only makes use of the intrinsic structure of the manifold.

### 3.1    GEOMETRICAL FEATURES & GAUGE TRANSFORMATIONS

The feature spaces in Gauge CNNs are modelled as fields $f$ over a manifold $M$. For example, the input data could be a vector field of wind directions on earth, or a scalar field of intensity values on the plane (a grayscale image), or a field of diffusion tensors on $\mathbb{R}^3$. We will refer to such quantities (scalars, vectors, tensors, and others) as *geometrical features* and speak of a field of geometrical features or geometrical feature field. Even if the input data consists only of scalars, one might

want to use other kinds of fields for the internal representation learned by the network, so we will describe Gauge CNNs for general fields. Because we are primarily interested in the 2-sphere $S^2$, we will specialize to the case of $d = 2$ dimensional Riemannian manifolds.

In computer science it is common to think of a vector or tensor as a list or array of numbers, but from a physical or mathematical perspective these are geometrical quantities that exist independent of a coordinatization / choice of basis. To represent a geometrical feature numerically however, we need to choose a frame for the tangent space $T_pM$ at each position $p \in M$. A smooth choice of frame is also known as a *gauge*. Mathematically, a gauge can be defined as a smoothly parameterized set of linear maps $w_p : \mathbb{R}^d \to T_pM$ (see Fig. 1).

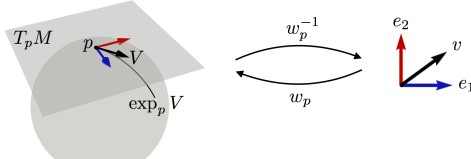

Since the choice of gauge is arbitrary, we should consider what happens to the coefficients of geometric features if we change it (i.e. apply a *gauge transformation*). Consider first the coefficients $f(p) = v$ of a tangent vector $V$ in the tangent space $T_pM$ at $p \in M$, expressed as a pair of numbers $v = (v_1, v_2)$ relative to an *orthogonal* frame $(w_p(e_1), w_p(e_2))$ at $p$. If we rotate the frame at $p$ by $r \in \mathrm{SO}(2)$, i.e. $w_p \mapsto w_p \circ r$, then the coefficient vector transforms as $v \mapsto r^{-1}v$. The vector itself is an abstract geometrical quantity, invariant to gauge transformations: $V = (w_p r)r^{-1}v = w_p v$.

Figure 1: The exponential map and the gauge $w_p : \mathbb{R}^2 \to T_pM$. The exponential map takes a tangent vector $V \in T_pM$ and follows the geodesic starting at $p$ with speed $\|V\|$ for one unit of time, to arrive at $q_v = \exp_p V \in M$. Figure courtesy of Cohen et al. (2019).

Note that we are free to change the gauge not just at one point, but at all positions simultaneously in an arbitrary (smooth) way. However, since we will only want to work with right-handed orthogonal frames, we only need to consider rotations of the frames. Thus, for our purposes we can define a gauge transformation as a smoothly varying choice of rotation $r_p \in \mathrm{SO}(2)$.

Beyond scalars (which are invariant to gauge transformations) and vectors (which transform like $f(p) \mapsto r_p^{-1}f(p)$), we will want to consider more general kinds of geometrical features. A $(2, 0)$-tensor, for instance, is (a linear combination of) tensor products $V \otimes W$ of vectors $V, W \in T_pM$. Given a frame, such a tensor is represented as a $d \times d$ matrix. Under a change of frame, a matrix $f(p)$ will transform like $f(p) \mapsto r_p f(p) r_p^{-1}$. We can also flatten the matrix into a $d^2$-dimensional coordinate vector $f(p)$, and write the transformation as $f(p) \mapsto (r_p \otimes r_p)f(p)$, where $r_p \otimes r_p$ is the Kronecker product.

The tensor product $\rho(r) = r \otimes r$ is an example of a group representation. This is a map $\rho : G \to \mathrm{GL}(C, \mathbb{R})$ taking each element $r$ of $G$ (the rotation group $\mathrm{SO}(2)$ in our case) to an invertible matrix $\rho(r)$ that acts on an $C$-dimensional feature vector. To be called a representation, it has to satisfy $\rho(rr') = \rho(r)\rho(r')$, which is easily checked for the tensor / Kronecker product.

Thus, we can generalize to geometric feature fields that transform like $f(p) \mapsto \rho(r_p^{-1})f(p)$ under gauge transformations, for any group representation $\rho$ of $\mathrm{SO}(2)$. We will refer to such fields as a $\rho$-field or a field of type $\rho$. In a gauge equivariant CNN, one chooses for each feature space of the network such a representation $\rho$ that determines the kind of features learned by that layer. The network is constructed such that a gauge transformation applied to the input will result in a corresponding gauge transformation in each feature space. The dimension $C$ of $\rho$ is equal to what is normally called the number of channels in the feature space. Typically, one would choose $\rho$ to be block-diagonal, containing for instance a number of scalar fields ($1 \times 1$ blocks $\rho_i(r) = 1$), a number of vector fields, etc. The number of copies of each type of feature is called its multiplicity.

## 3.2 GAUGE EQUIVARIANT CONVOLUTION

For each layer of the network, we want to interpret both the input and output as fields of geometrical features. If we apply a gauge transformation, the input coefficients change ($f(p) \mapsto \rho(r_p^{-1})f(p)$), and we want the same to happen to the output (*gauge equivariance*), so that we may interpret it as the coefficients of a geometrical quantity relative to a gauge. In this section we will define a convolution-like operation that has this property.

The classical convolution operation involves summing or integrating the product of a filter and the input signal over a local region. If we want to generalize this to fields on a manifold, we run into a difficulty: the geometric feature $f(p)$ and $f(q)$ at different points $p$ and $q$ in $M$ live in a different vector space, and so we cannot directly compare them or add them up. For instance, if we have two tangent vectors $V \in T_pM$ and $W \in T_qM$, how can we say that they are "the same" or add them up? Having chosen a frame for $T_pM$ and $T_qM$ we could add the coordinate vectors $v \in \mathbb{R}^2$ and $w \in \mathbb{R}^2$, but because we can change the gauge of both tangent spaces independently, the result is not the coefficient vector of any invariant geometrical quantity.

The solution is to apply *parallel transport* to the feature vectors before adding them up. Given a curve from $q$ to $p$, we can transport a vector $W \in T_qM$ to $T_pM$ by applying a rotation $r_{p\leftarrow q} \in \mathrm{SO}(2)$ to its coefficient vector $w$. Since we can interpret $r_{p\leftarrow q}w$ as a vector in $T_pM$, the quantity $v + r_{p\leftarrow q}w$ is well-defined. In general, for other kinds of geometrical features, parallel transport acts via $\rho$, i.e. we can add $v + \rho(r_{p\leftarrow q})w$. We will use this in the definition of the gauge equivariant convolution.

Following Masci et al. (2015), we parameterize a local neighborhood around $p \in M$ by the tangent plane $T_pM \simeq \mathbb{R}^2$ via exponential map (see Fig. 1). That is, we index nearby points $q$ by tangent vectors using the exponential map, by defining $q_v = \exp_p w_p v$ for $v \in \mathbb{R}^2$ ("Riemannian normal coordinates"). The convolution is then defined by transporting for each nearby point $q_v$ the feature vector $f(q_v)$ to $p$ by computing $\rho(r_{p\leftarrow q_v})f(q_v)$, transforming the resulting features at $p$ using a learned kernel $K : \mathbb{R}^2 \to \mathbb{R}^{C_{\mathrm{out}} \times C_{\mathrm{in}}}$, and integrating the result over the support of $K$ in $\mathbb{R}^2$:

$$\psi \star f(p) = \int_{\mathbb{R}^2} K(v)\rho_{\mathrm{in}}(r_{p\leftarrow q_v})f(q_v)dv. \tag{1}$$

As shown by Cohen et al. (2019), this operation is gauge equivariant if and only if $K(v)$ satisfies

$$K(r^{-1}v) = \rho_{\mathrm{out}}(r^{-1})K(v)\rho_{\mathrm{in}}(r). \tag{2}$$

In section 4.2 we show how we can parameterize such a kernel via rotational weight-sharing.

### 3.3 Equivariance to $\mathrm{SO}(3)$

In addition to gauge equivariance, $\mathrm{SO}(3)$ equivariance is a desirable property for a Spherical CNN (Cohen et al., 2018). This means that if we apply a 3D rotation to the input of the network, the output is also rotated. In this section we show that the gauge equivariant convolution as defined is also equivariant to $\mathrm{SO}(3)$.

Consider a local patch on the sphere (e.g. the support of the kernel), and the signal defined there. When we rotate the sphere, the patch is moved to another place, and it may change its orientation. Moving the patch is not a problem: at the new position we apply the same kernel $K$, so one expects that the convolution result at the new position equals the convolution result of the original signal at the old position. However, since the orientation of the kernel is determined by the gauge (which is arbitrary but fixed) and because we can arbitrarily change the orientation of the patch by rotating around its center, the kernel and the patch may be matched in a different relative orientation after applying the rotation. Fortunately, because the kernel satisfies Eq. 2, the result will be equivalent up to a gauge transformation acting by $\rho_{\mathrm{out}}$, and so we have $\mathrm{SO}(3)$ equivariance.

Thus, in the continuous theory, the gauge equivariant convolution is also $\mathrm{SO}(3)$ equivariant. However, as we will see, making sure that this holds in a discrete implementation is not entirely trivial.

## 4 Discrete Implementation of Spherical Gauge CNNs

The theory covered so far tells us how, mathematically, we can define a convolution-like operation that is gauge equivariant. However, it does not tell us exactly how to implement it on a computer, in order to process discretely sampled signals on a manifold. How we do this exactly can have a large effect on the efficiency and numerical accuracy of the method.

A signal is represented as a list of values $f_i = f(p_i)$ associated with a finite number of points $p_i \in \mathcal{V} \subset S^2$ (see Sec. 4.1). We assume that the kernel $K(v)$ has local support, so $K(v) = 0$ whenever $\|v\| > R$ for some radius $R$. Equivalently, we can say that $q \in S^2$ only contributes to the

convolution result at $p \in S^2$ if the geodesic distance between $p$ and $q$ is smaller than $R$. Accordingly, we define the set of neighbors $\mathcal{N}(p)$ of $p$ as the set of points $q$ within radius $R$ from $p$.

A simple way of discretizing the gauge convolution (Eq. 1) is to replace the integral over $\mathbb{R}^2$ (identified with $T_pM$) by a sum over neighbors of $p$. Each neighbor can be associated with a tangent vector via the logarithmic map: $v_{pq} = \log_p q$. This yields the following approximation:

$$\psi \star f(p) = \sum_{q \in \mathcal{N}(p)} K(v_{pq})\rho_{\text{in}}(r_{p\leftarrow q})f(q) \tag{3}$$

At this point it is worth comparing Eq. 3 to the simplest form of message-passing based graph convolution (Kipf & Welling, 2017; Gilmer et al., 2017). In both cases, the result of convolution is computed as a sum of *messages* comming from neighbors. In the case of graph CNNs, these messages are computed as $Af(q)$, where $A$ is a weight matrix that is shared by all neighbors. So graph convolutions use isotropic filters that cannot distinguish where the message comes from.

The gauge convolution sums messages of the form $K(v_{pq})\rho_{\text{in}}(r_{p\leftarrow q})f(q)$. Thus, the feature vectors $f(q)$ of neighbors $q$ are transformed in a way that depends i) on the intrinsic geometry of the manifold via $r_{p\leftarrow q}$ and $v_{pq}$, and ii) by a non-isotropic (but gauge-equivariant) learnable kernel $K(v_{pq})$. We can thus think of gauge convolutions (implemented in this way) as a more powerful version of graph convolution that leverages the additional topological and metric structure of a Riemannian manifold to process geometrical data in a more flexible manner.

The discrete gauge convolution is computed in a few steps, some of which are done during precomputation and some during the forward pass: i) the logarithmic map $v_{pq} = \log_p q$, ii) the parallel transporter $r_{p\leftarrow q}$, iii) the construction / parameterization of the kernel $K(v)$, iv) the linear contraction of the kernel and the signal.

Steps (i) and (ii) can be complicated for a general manifold or mesh, but for the sphere are easily and exactly computable in closed form with standard operations, which are detailed in Appendix B. We will discuss our method for selecting the finite number of points on the sphere, as well as steps (iii) and (iv) in the following sections.

## 4.1 THE ICOSPHERE GRID

Even though our method uses the exact geometry of the sphere, a grid of points on the sphere must nevertheless be chosen on which the features will live. The sphere does not admit perfectly symmetrical and homogeneous high resolution grids. A grid that is fairly homogeneous and has been used successfully in Spherical CNNs before is what we call the icosphere grid (Jiang et al., 2019; Liu et al., 2019; Cohen et al., 2019). The icosphere grid can be computed at different levels of resolution. The lowest resolution $s = 0$ has as points $p_i$ the 12 corners of the icosahedron. Higher resolutions are obtained by repeated subdivision of the triangular faces of the icosahedron into 4 sub-triangles, followed by a projection of all points to the sphere. The result is a grid $\mathcal{H}_s$ with $5 \times 2^{2s+1} + 2$ points at subdivision level $s$. We emphasize that our method is not in any way tailored to this grid, and other options such as HEALPix (Górski et al., 2005) could easily be substituted.

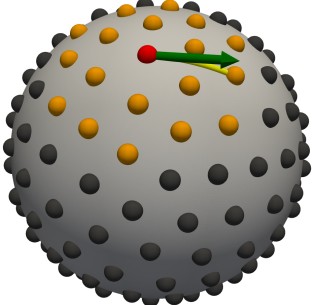

Figure 2: Icosphere with $s = 2$. The orange points are the 2-ring neighbours of the red point. The green arrow is the logarithmic map corresponding to the yellow path.

## 4.2 PARAMETERIZATION OF THE KERNEL

The kernel $K(v)$ is defined as a continuous matrix-valued function of $\mathbb{R}^2$ that satisfies the kernel constraint (Eq. 2). In a classical CNN, where we are dealing with a homogeneous grid of pixels in $\mathbb{R}^2$, we can define a small (e.g. $3 \times 3$) set of neighboring pixels $\mathcal{N}(p) = \{p + v^{(i)}\}_i$ so that we only ever need to evaluate the kernel at a small number (e.g. 9) of points $v^{(i)}$. This results in a parameterization of $K$ as an array with $C_{\text{out}} \times C_{\text{in}} \times 3 \times 3$ learnable coefficients.

On the sphere there are no perfectly homogeneous grids, so depending on the point $p \in \mathcal{V}$ where we are evaluating the convolution $\psi \star f$, the neighborhood structure $\mathcal{N}(p)$ may look quite different.

Hence, the points $v_{pq} \in \mathbb{R}^2$ where we need to evaluate $K$ will differ as well. For this reason, we parameterize $K$ as a linear combination of analytically determined continuous basis kernels. The linear coefficients will be learned.

We assume that $\rho_{\text{in}}$ and $\rho_{\text{out}}$ are block-diagonal with irreducible representations (irreps) as blocks (any $\text{SO}(2)$ representation can be brought to this form by a change of basis). In this case the kernel takes on a block structure as well, with each block corresponding to a particular input/output irrep (Cohen & Welling, 2017), with irreps labelled by integer frequency $n \geq 0$ (Worrall et al., 2017) (see Appendix D). So we will focus on the case where both input and output representation consist of a single irrep, and construct the full kernel block-wise as described in Appendix A.

As derived in Appendix D, the analytical solutions to Eq. 2 can be split in a independent radial part and angular part. The solutions for the angular part $K(\theta)$ are shown in Table 1, while the radial part is unconstrained. So if a set of radial functions $\{R_a(r)\}$ are chosen, and $\{K_b(\theta)\}$ is the complete set of angular solutions, the parameterized kernel is: $K(r, \theta) = \sum_{ab} w_{ab} R_l(a) K_b(\theta)$ for weights $w$. Hereafter, we denote one such solution as $K_i$, so that the parameterized kernel is $\sum_i w_i K_i$. The number of basis-kernels is called `num_basis`.

| $\rho_{\text{in}} \to \rho_{\text{out}}$ | Linearly independent solutions for $K(\theta)$ |
|---|---|
| $\rho_0 \to \rho_0$ | $1$ |
| $\rho_n \to \rho_0$ | $(\cos n\theta \quad \sin n\theta), (\sin n\theta \quad -\cos n\theta)$ |
| $\rho_0 \to \rho_m$ | $\begin{pmatrix} \cos m\theta \\ \sin m\theta \end{pmatrix}, \begin{pmatrix} \sin m\theta \\ -\cos m\theta \end{pmatrix}$ |
| $\rho_n \to \rho_m$ | $\begin{pmatrix} c_- & -s_- \\ s_- & c_- \end{pmatrix}, \begin{pmatrix} s_- & c_- \\ -c_- & s_- \end{pmatrix}, \begin{pmatrix} c_+ & s_+ \\ s_+ & -c_+ \end{pmatrix}, \begin{pmatrix} -s_+ & c_+ \\ c_+ & s_+ \end{pmatrix}$ |

Table 1: Solutions to the angular kernel constraint for kernels that map from $\rho_n$ to $\rho_m$. We denote $c_\pm = cos(m \pm n)\theta, s_\pm = \sin(m \pm n)\theta$.

Since the geometry and grid are fixed, we can precompute the basis kernels evaluated at all required points. That is, for each $p \in \mathcal{V}$ and $q \in \mathcal{N}(p)$ we evaluate each basis kernel contracted with the input representation $K_i(v_{pq})\rho_{\text{in}}(r_{p \leftarrow q})$ where $r_{p \leftarrow q}$ and $v_{pq}$ are as computed in Section B.1 and B.2. The result of this precomputation is an array of shape `num_basis` × `num_v` × `num_neigh` × `c_out` × `c_in`, where `c_in` and `c_out` are the dimensionality of $\rho_{\text{in}}$ and $\rho_{\text{out}}$ and also the number of channels of the input and output signals.

### 4.3 COMPUTING THE CONVOLUTION

Having computed the basis kernels at each $v_{pq}$, we can compute the discretized gauge convolution (Eq. 3) as a linear contraction. This is done in two steps. Initially, we expand the signal $f(p)$, which has shape `num_v` × `c_in`, to $\hat{f}$ of shape `num_v` × `num_neigh` × `c_in`. This is done so that $\hat{f}_{pq}$ is the value of the signal at the $q$-th neighbor of $p$.

Subsequently, we contract the signal $\hat{f}$ with basis kernels $K_i(v_{pq})\rho_{\text{in}}(r_{p \leftarrow q})$ and weights $w_i$ to obtain the convolution result $\psi \star f$ of shape `num_v` × `c_out`. Since a basis-kernel $K_i$ only acts on one in/out irrep pair, it is mostly zero. In Appendix A, we detail how this block-sparsity can get exploited for computational efficiency. We note that $\psi \star f$ can easily be computed for a subset $\overline{\mathcal{V}} \subset \mathcal{V}$.

### 4.4 NONLINEARITIES

For the network to be gauge equivariant, every layer should be gauge equivariant, including nonlinearities. Irrep features do not commute with pointwise nonlinearities (Worrall et al., 2017; Thomas et al., 2018; Weiler et al., 2018; Kondor et al., 2018). However, we can perform a basis transformation to a basis in which pointwise non linearities are approximately gauge equivariant. Afterwards, we transform the basis back to the irreps.

For simplicity, we assume that the representation is $U$ copies of $\rho_0 \oplus \rho_1 \oplus ... \oplus \rho_M$. One such copy can be treated as the discrete Fourier modes of a circular signal with band limit $M$. An inverse Discrete Fourier Transform (DFT) matrix can map these modes to $N$ spatial samples. Under a gauge transformation of a multiple of $2\pi/N$, the samples are cyclically shifted. The resulting representation can thus be called a regular representation and hence our procedure a `RegularNonlinearity`. Non linearities that act pointwise on these samples, such as the ReLU, commute with such gauge transformations. The procedure is however only approximately gauge equivariant under gauge transformations of angles that are not multiples of $2\pi/N$. Nevertheless, we prove in Appendix E that in

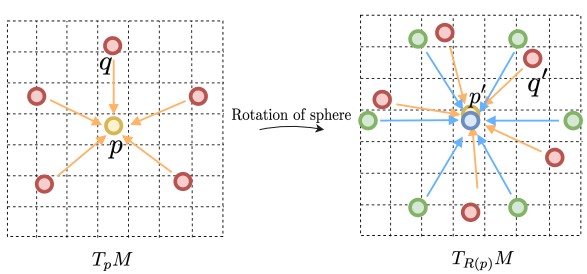 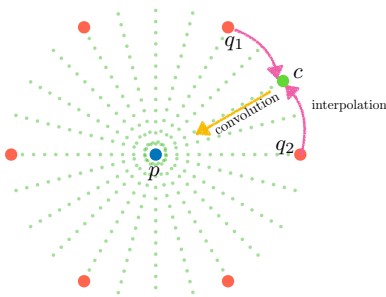

(a) Any spherical grid is irregular. For example, when the the sphere is rotated, point $p$, with 5 nearby neighbors, can be mapped to a grid point $p'$, which has 6 neighbors. As a result, the naive method may fail to be rotation equivariant.

(b) For each quadrature point $c \in \mathcal{I}$ (green), the signal interpolated from the neighbors $\mathcal{N}(p)$ (red). The convolution is equal to the sum over quadrature points, but we only sum over neighbors during the forward pass.

Figure 3: Quadrature Integration

the limit $N \to \infty$, exact equivariance is restored and we provide a finite error bound, which can help selecting $N$.

## 5  QUADRATURE INTEGRATION

The implementation described in the previous section satisfies gauge equivariance exactly. In addition to gauge symmetry, the sphere has global rotational / $SO(3)$ symmetry, and like previous methods we would like our method to be equivariant to 3D rotations as well. In the continuous case, gauge equivariance implies $SO(3)$ equivariance, but the scheme above is not exactly $SO(3)$ equivariant due to discretization (see Figure 3a). In this section we describe a numerical method that improves $SO(3)$ equivariance at no additional computational cost during training.

Our general approach for computing $\psi \star f(p)$ will be to interpolate the sample values at $\mathcal{N}(p)$ to obtain a continuous function on $\mathbb{R}^2$, and then use quadrature integration to get a more precise value for the integral. Quadrature is a general numerical technique for approximating integrals with finite sums. For some region $A$ and function $g$, the integral $\int_A g(x)dx$ can be approximated by $\sum_{x \in \mathcal{I}} \omega_x g(x)$, where $\mathcal{I} \subset A$ is some finite set of *quadrature points*, each with a weight $\omega_x$. The goal is selecting $\mathcal{I}$ and $\omega_x$ such that the approximation is accurate (or even exact), for functions $g$ satisfying some regularity assumptions (e.g. being band-limited). In our case, the region $A$ is a disk with as radius the support radius $R$ of the kernel. The quadrature rule is described in Appendix C.

The signal at $c \in \mathcal{I}$ is inferred from the signals at $\mathcal{N}(p)$ by interpolation:

$$\tilde{f}_p(c) = \frac{1}{Z(p,c)} \sum_{q \in \mathcal{N}(p)} k(c,q)\rho_{\text{in}}(g_{p \leftarrow q})f(q) \tag{4}$$

where $k(c,q) = \exp\left(-||c - \log_p(q)||^2/\sigma^2\right)$ is a Gaussian kernel with scale $\sigma$, measuring distance between $c$ and $q$ in the tangent space, and $Z(p,c) = \sum_{q \in \mathcal{N}(p)} k(c,q)$ is a normalizing constant.

We can now compute the integral over $\mathbb{R}^2$ by quadrature integration:

$$\psi \star f(p) = \sum_{c \in \mathcal{I}} \omega_c K(c)\tilde{f}_p(c) \tag{5}$$

The convolution Eq. 5 sums over a homogenized neighborhood and is thus more equivariant to rotations of the sphere. See Figure 3b for an illustration. Equivariance improves if a large number of quadrature points are used, which increases the computational cost. Luckily, since the composition

of linear operations is linear, we simplify:

$$\psi \star f(p) = \sum_{c \in \mathcal{I}_p} \omega_c K(c) \tilde{f}_p(c) = \sum_{q \in \mathcal{N}(p)} \sum_{c \in \mathcal{I}_p} \frac{\omega_c k(c,q)}{Z(p,c)} K(c) \rho_{\text{in}}(g_{p \leftarrow q}) f(q) = \sum_{q \in \mathcal{N}(p)} \hat{K}(p,q) f(q)$$

(6)

for a new kernel $\hat{K}(p,q) = \sum_{c \in \mathcal{I}_p} \frac{\omega_c k(c,q)}{Z(p,c)} K(c) \rho_{\text{in}}(g_{p \leftarrow q})$. The new kernel $\hat{K}$ can be pre-computed once, so that the convolution during run-time involves only a sum over the neighbors, just as in the naive convolution (Eq. 3). The interpolation thus does not affect computational cost.

## 6 EXPERIMENTS

In this section, we present experimental results on three benchmark problems. Throughout this section, we will refer to the gauge equivariant convolutions described in 4.3 as `IrrepConvolution` and similarly refer to tailored batch normalization and nonlinearities as `IrrepBatchNorm` (normalizing irreps instead of channels) and `RegularNonlinearity` as we name in our implementation. An `IrrepConvolution` and other operations are defined by specifying the number of input and output multiplicities for each irrep order, in analogy to number of channels in regular CNNs. In addition, we need to specify the *types* of irreps by their angular frequency. We do that by `max_freq` argument which determines the cutoff frequency. We will describe $\rho$ by a parameter `max_freq` and multiplicity, and include each frequency $0, \ldots, $ `max_freq` with the same multiplicity.

### 6.1 SPHERICAL MNIST

We perform a series of experiments on a toy dataset to validate the equivariance and generalization properties of our gauge-equivariant spherical CNN. Specifically, we compare our method with different filter sizes (1-ring, 2-ring) with and without interpolation, to three prior methods, as well as an isotropic baseline. The isotropic baseline uses only scalar features and isotropic kernels, but is similar in number of parameters to our model.

We follow a setup similar to that of Cohen et al. (2019) and generate Spherical MNIST in randomly Rotated (R) and Non-rotated (N) conditions. For all experiments, we represent the signals on an icospherical grid at level $s = 4$. For network architecture and training procedure, see Appendix F.1.

We present our results in Table 2 on three configurations of the Spherical MNIST dataset: N/N, N/R and R/R, where X/Y means we train on X and test on Y. In addition, we run an ablation study to test the effect of quadrature integration on equivariance. Whether interpolation is used or not, our method performs similarly to two relevant spherical CNN baselines in terms of classification score on N/N. We observe that our method automatically generalizes to 3D rotations without data augmentation and thus compares favorably against IcoCNN in the N/R condition. However, in the ablation study, we observe that our method's generalization capability against 3D rotations drops in the absence of interpolation, particularly when smaller (1-ring) filters are used. This result shows the usefulness of quadrature integration when using small filters.

Our model outperforms FFS2CNN in all categories. Furthermore, it performs better than the isotropic baseline, which illustrates the importance of anisotropic filters and non-scalar features for the MNIST dataset. This is to be expected, as detection of line direction is naturally expressed in anisotropic filters.

As a final note, our results confirm the theoretical result that gauge equivariance implies SO(3)-equivariance (Sec. 3.3).

### 6.2 CLIMATE PATTERN SEGMENTATION

We put our model to the test on a climate pattern segmentation dataset proposed by Mudigonda et al. (2017). The dataset is collection of simulated climate variables over 20 years from the Community Atmosphere Model CAM-5.0 (Neale et al., 2010). The task is to segment out regions of Atmospheric Rivers (AR) and Tropical Cyclones (TC) from background (BG). We use the preprocessed data

---

[1]Results from (Cohen et al., 2019).

| Method | N/N | N/R | R/R | Time / epoch | Complexity |
|---|---|---|---|---|---|
| S2CNN (Cohen et al., 2018)[1] | 99.38 | 99.38 | 99.12 | 380s | $\mathcal{O}(N \log N)$ |
| IcoCNN (Cohen et al., 2019) | 99.43 | 69.99 | 99.31 | 72s | $\mathcal{O}(N)$ |
| FFS2CNN (Kondor et al., 2018) | 96.4 | 96 | 96.6 | | $\mathcal{O}(N)$ |
| Isotropic (2-ring, interpolation) | 98.27 | 95.31 | 97.18 | 87s | $\mathcal{O}(N)$ |
| Ours (1-ring, interpolation) | 99.47 | 97.65 | 99.24 | 94s | $\mathcal{O}(N)$ |
| Ours (2-ring, interpolation) | 99.51 | 99.32 | 99.43 | 284s | $\mathcal{O}(N)$ |
| Ours (1-ring, no interpolation) | 99.30 | 91.60 | 99.17 | 94s | $\mathcal{O}(N)$ |
| Ours (2-ring, no interpolation) | 99.45 | 98.48 | 99.28 | 284s | $\mathcal{O}(N)$ |

Table 2: Spherical MNIST results

released by Jiang et al. (2019) as is. The data consist of 16 channels sampled on an icospherical grid at level $s = 5$ (i.e. 10242 vertices).

For this experiment, we use a residual U-Net architecture. Our residual blocks contain two blocks of [`IrrepConvolution`, `IrrepBatchNorm`, `RegularNonlinearity`] and a skip connection. We set the multiplicity to 2 at the input resolution $s = 5$ and fix `max_freq` $= 2$ throughout. The rest of the encoder stream progressively reduces the grid resolution down to $s = 0$ while doubling the multiplicities (and thus the number of channels) every time the resolution is decreased, resulting in 196 channels. The decoder stream is mirror-symmetrical to the encoder, replacing downsampling layers with upsampling layers. We use 2 residual blocks at each resolution level. The network is trained for 60 epochs with an initial learning rate of 0.01 which is reduced by a factor of 0.4 every 30 epochs. Batch size is 64. We use a weighted cross-entropy loss, due to class imbalance following the setup of Jiang et al. (2019), with Adam optimizer (Kingma & Ba, 2015).

In Table 3, we present our results. Our model compares favorably against the two spherical CNNs that have reported results on this dataset (Jiang et al., 2019; Zhang et al., 2019). Both methods operate on the same icospherical grid as our method. Our method is on par with the Icosahedral CNN of Cohen et al. (2019) on accuracy while significantly outperforming on the more relevant mAP metric. We attribute this gain to the fact that our model reflects the geometry of the signal more faithfully than IcoCNN which operates on an approximated sphere, namely, icosahedron.

### 6.3 OMNIDIRECTIONAL SEMANTIC SEGMENTATION

Next, we run our model in omnidirectional semantic segmentation task. For this experiment, we use Stanford 2D3DS dataset which consists of 1413 omnidirectional RGBD images acquired in 6 different locations with labels of 13 semantic categories. We are following the experimental setups in (Jiang et al., 2019; Cohen et al., 2019) and report averages over 3 cross-validation splits.

We use a residual U-Net similar to what is described in 6.2. The only difference is that we use 4 residual blocks per resolution instead of 2. We train the network using weighted cross-entropy loss and Adam optimizer for 100 epochs. We set batch size to 16 and initial learning rate to 0.01. Learning rate is reduced by a factor of 0.7 every 20 epochs.

We report our results in Table 4. We observe that our rotation equivariant network performs similar to the orientation-aware network of Zhang et al. (2019) while outperforming IcoCNN which is only equivariant up to discrete subgroup, A5 (i.e. icosahedral group), of SO(3). We note that 2D3DS is acquired with a preferred camera orientation rendering an additional challenge to our equivariant model which assumes no preferred orientation. Yet, the comparison against orientation-aware model of Zhang et al. (2019) reflects the expressive capability of ours despite the disadvantage.

| Method | BG | TC | AR | Average (%) | mAP (%) |
|---|---|---|---|---|---|
| Jiang et al. (2019) | 97.0 | 94.0 | 93.0 | 94.7 | N/A |
| Zhang et al. (2019) | 97.3 | 96.3 | 97.5 | 97.0 | 55.5 |
| Cohen et al. (2019) | **97.4** | **97.9** | 97.3 | **97.7** | 75.9 |
| Ours | 97.0 | 96.7 | **97.6** | 97.1 | **80.6** |

Table 3: Results on Climate Pattern Segmentation

| Method | Mean Acc. (%) | Mean IoU (%) |
|---|---|---|
| Jiang et al. (2019) | 54.7 | 38.3 |
| Cohen et al. (2019) | 55.9 | 39.4 |
| Zhang et al. (2019) | **58.6** | **43.3** |
| Ours | 58.2 | 39.7 |

Table 4: Results on Stanford 2D3DS

## 6.4 ATOMIZATION ENERGY PREDICTION

In this experiment, we test our method on molecular energy regression task given the molecular geometry and charges associated with each atom. To this end, we use QM7 dataset (Blum & Reymond, 2009; Rupp et al., 2012) which consists of molecules with varying number of atoms $N$ ($\leq 23$). The molecules considered in this dataset are made up of $T = 5$ different elements (i.e. H, C, N, O and S). Each molecule is defined by a tuple $(p_i, z_i)$ with $i$ indexing the atoms. We follow the setup for signal representation outlined in Cohen et al. (2018) and represent geometry of molecules with spheres, $s_i$, of constant radius centered at atomic positions, $p_i$, in a careful arrangement such that no two spheres intersect in a molecule. Finally, each atom $i$ is associated with $T$-channel spherical potential functions describing atomic interactions within the molecule with $U_z(x)_{x \in s_i} = \sum_{j \neq i, z_j = z} (z_i z)/|x - p_i|$.

We use a similar architecture to that of Cohen et al. (2018) in terms of parameters. The details of network and training is available in Appendix F.2. Our method achieves the lowest RMSE error in comparison to the spherical baselines as shown in Table 5.

| Model | RMS error |
|---|---|
| MLP/Random CM (Montavon et al., 2012) | 5.96 |
| FFS2CNN (Kondor et al., 2018) | 7.97 |
| S2CNN (Cohen et al., 2018)[1] | 5.13 |
| Ours | **4.99** |

Table 5: Results on QM7 Atomization Energy Prediction task.

| | $\sigma = 0.0$ | $\sigma = 0.5$ | $\sigma = 1.0$ | $\sigma = 1.5$ | $\sigma = 2.0$ |
|---|---|---|---|---|---|
| DeepSphere (FCN) | 100.0 | 99.79 | 97.70 | 92.52 | 86.84 |
| DeepSphere (CNN) | 100.0 | 99.71 | 97.18 | 91.25 | 83.61 |
| Ours | 99.27 | 98.23 | 96.29 | 91.80 | 84.90 |

Table 6: Results on Cosmology dataset. Split level 4.

## 6.5 COSMOLOGICAL MODEL CLASSIFICATION

We test our model on cosmological model classification task which is an interesting test bed to demonstrate our algorithm's flexibility in admitted sampling scheme and scalability aspects. To this end, we use a dataset released by Perraudin et al. (2018). This dataset consists of cosmological convergence maps which are generated by whole-sky N-body simulations for two different parameter settings of $\Lambda$CDM cosmological model. Both cosmological models are simulated 30 times. For testing, 20 simulations are held out. The remaining 40 simulations are further split into training and validation sets of size 32 and 8, respectively. The signals are represented on HEALPix grid. We refer the reader to the original article for more information.

We follow the experimental setup described in (Perraudin et al., 2018) for evaluation. We split the maps into $12 \times o^2$ patches where $o = 4$ indexes the hierarchical level of the patches in HEALPix sampling scheme. We then obtain signals of dimensionality $(N_{side}/o)^2$ where $N_{side}$, i.e. number of pixels on one side of HEALPix patch, is set to $2^{10}$. During training, we apply additive Gaussian noise with varying standard deviation, $\sigma_{noise} \in \{0, 0.5, 1, 1.5, 2\}$ and report results separately on each setting. The details of network architecture and training are available in Appendix F.3. The results in Table 6 show that our model performs similarly to the graph-based approach.

## 7 CONCLUSION

We have introduced the gauge equivariant spherical CNN, which is simultaneously efficient, numerically accurate, flexible with respect to the pixel grid used, and which can in principle be applied to data that is defined on local parts of the sphere only. Additionally, it is the first implementation of the theory of gauge CNNs for a manifold that is not locally flat. Our experimental results show that the method is accurate and achieves strong performance on various benchmark problems. Moreover, our method can be adapted to work on meshes, and we plan to work on this in the future.

---

[1]Results from running code in repository `https://github.com/jonas-koehler/s2cnn`.

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

## A IMPLEMENTATION OUTLINE

This section outlines how the Gauge equivariant convolution is implemented. PyTorch notation is used and some details are omitted. During pre-computation, the kernel tensor is constructed, which is used during runtime to perform the convolution.

A naive implementation would construct for each basis kernel a large matrix mapping from all input channels to all output channels. The parametrized kernel is then obtained from contracting these with a weight vector, which has a weight for each basis kernel. However, this stack of basis kernels would be very large for two reasons. Firstly, since the grid is irregular, a separate copy needs to be stored for each point on the sphere. Secondly, it is common to use irreps of low orders, but use many copies of those (so with a *multiplicity* larger than one), each having the same set of basis kernels.

We prevent creation of this large stack of basis kernels by exploiting the fact that each basis kernel is a very sparse linear transformation, as it only has one irrep as input and one as output, making it block-sparse. This allows us to group the basis kernels mapping from and to the same irrep. So we can exploit the block-sparsity and also prevent allocating a kernel basis for each multiplicity.

The kernels between two irreps can be constructed as follows. Arguments are the orders of the input and output irreps, a set of radial activation functions, and for each point $p$ and neighbor $q$, the radius and angle of the log map from $p$ to $q$ in the gauge at $p$, the parallel transport angle.

```python
def build_kernel(order_in, order_out, radial_functions, log_map_r,
                 log_map_angle, transport, mask):
    """Last four arguments are [num_v, num_neigh].
    Return [num_basis, num_v, num_neigh, dim_out, dim_in]."""
    a_in = order_in * (log_map_angle - transport)
    a_out = order_out * log_map_angle
    cos_min = np.cos(a_out-a_in)
    cos_plus = np.cos(a_out+a_in)
    sin_min = np.sin(a_out-a_in)
    sin_plus = np.sin(a_out+a_in)
    if order_in == 0 or order_out == 0:
        ...
    else:
        angular_part = np.array([
```

```
                    [[cos_min, -sin_min],    [sin_min, cos_min]],
                    [[sin_min, cos_min],     [-cos_min, sin_min]],
                    [[cos_plus, sin_plus],   [sin_plus, -cos_plus]],
                    [[-sin_plus, cos_plus],  [cos_plus, sin_plus]],
                ])
            radial_part = np.array([f(log_map_r) for f in radial_functions])
            kernel = np.einsum(
                'rpq,pq,bijpq->brpqij', radial_part, mask, angular_part)
            return kernel.reshape((-1, *kernel.shape[2:]))
```

Given such kernels for all irrep orders, we subsequently can define the gauge equivariant convolution.

Basically, we iterate over each irrep in the in dimension and each irrep in the out dimension. For each block, we have $m_{in} * dim(order_{in})$ input dimensions and $m_{out} * dim(order_{out})$ output dimensions.

```
1   x = ...   # [num_batch, num_v, c_in]
2   # The layout of the channels is:
3   # cat(flatten((irrep order, irrep multiplicity, irrep dimension)))
4   neighbors = ... # [num_v, num_neigh]
5   reps_in = [(0, 4), (1, 4)]   # 4 irreps of order 0, 4 of order 1
6   reps_out = [(0, 4), (1, 3)]  # 4 irreps of order 0, 3 of order 1
7   kernels = ...
8   # {(order_out, order_in): [num_basis, num_v, num_neigh, d_out, d_in]}
9
10  # One weight for each:
11  # in/out irrep * number of basis kernels * multiplicities in/out
12  weights = torch.randn(sum(
13      len(kernels[(order_out, order_in)]) * m_out * m_in
14      for order_out, m_out in reps_out
15      for order_in, m_in in reps_in))
16
17  # Neighborhood expansion
18  x_e = x[:, neighbors]
19  num_batch, num_v, num_neigh, c_in = x_e.shape
20
21  # Dimension of representation of order l
22  dim = lambda order: 1 if order == 0 else 2
23
24  c_out = sum(dim(l) * m for l, m in reps_out)
25  y = torch.zeros(num_batch, num_v, c_out)
26  y_idx = 0
27  w_idx = 0
28  # Iterate over in/out irreps. Concatenate over out irreps
29  for order_out, m_out in reps_out:
30      x_idx = 0
31      for order_in, m_in in reps_in:  # Sum over in irreps
32          k = kernels[(order_out, order_in)]
33          x_rep = x_e[:, :, :, x_idx:x_idx + m_in * dim(order_in)] \
34              .view(num_batch, num_v, num_neigh, m_in, dim(order_in))
35          w_rep = weights[w_idx:w_idx + len(k) * m_out * m_in] \
36              .view(len(k), m_out, m_in)
37          y[:, :, y_idx:y_idx + dim(order_out) * m_out] += torch.einsum(
38              'uvb,bpqij,npqvj->npui', w_rep, k, x_rep)
39          x_idx += m_in * dim(order_in)
40          w_idx += len(k) * m_out * m_in
41      y_idx += dim(order_out) * m_out
42  return y
```

# B  SPHERICAL GEOMETRY

To compute the logarithmic map at $p$ whose exponential map lands at $q$, and to compute the parallel transporter from $q$ to $p$ along a geodesic, we do the following.

## B.1  PARALLEL TRANSPORTERS

Since $r_{p \leftarrow q}$ is a planar rotation, it is determined by where it sends a single (non-zero) vector. We take the first basis vector $b_1^q = w_q((1,0))$ and express it in 3D Euclidean coordinates. We then rotate this vector by the angle $\angle(p,q) = \arccos \langle p, q \rangle$ between $p$ and $q$ around the axis $p \times q$ which is orthogonal to the $pq$ plane. The resulting vector lies in the tangent plane at $p$. Then $r_{p \leftarrow q}$ is determined as the angle between this vector and the first basis vector $b_1^p = w_p((1,0))$ in $T_p S^2$.

We precompute the transport angles for every point $p$ in the grid $\mathcal{V}$ and every $q \in \mathcal{N}(p)$. This results in an array of angles of size `num_v` $\times$ `num_neigh` where `num_v` $= |\mathcal{V}|$ and `num_neigh` $= \max_{p \in \mathcal{V}} |\mathcal{N}(p)|$ is the maximum neighborhood size. For nodes with a non-maximal number of neighbors, we pad with zeros.

## B.2  LOGARITHMIC MAPS

For each $p \in \mathcal{V}$ and $q \in \mathcal{N}(p)$ we need to compute $v_{pq} = \log_p q$, which is the vector in $T_p S^2$ that points in the direction of $q$ and has length equal to the geodesic distance between $p$ and $q$. One way to compute the log map is to project the 3 dimensional vector pointing from $p$ to $q$ to the tangent plane. This is done by subtracting from $q - p$ the component parallel to the $p$ vector, as the $p$ vector is orthogonal to the tangent plane at $p$. This produces a vector $\tilde{v} \propto (q - p) - \langle p, q - p \rangle p = q - \langle p, q \rangle p$ which has the right direction. Then one can scale the length of $\tilde{v}$ so that it matches the geodesic distance $d(p, q)$ (the arclength):

$$\log_p q = d(p, q) \frac{q - \langle p, q \rangle p}{\| q - \langle p, q \rangle p \|} \tag{7}$$

We express the result $v_{pq} = \log_p q$ in polar coordinates relative to the gauge at $p$. This gives two arrays `log_map_r` (the length / radial coordinate of $v$) and `log_map_angle` (the angular part of $v$, relative to the gauge at $p$). Both are shaped `num_v` $\times$ `num_neigh` as before. Since the geometry and grid are fixed, these arrays are computed only once before training.

# C  QUADRATURE

The goal of selecting the quadrature points and weights is to approximate an integral $\int_D g(x) dx$, where $D \subset \mathbb{R}^2$ is the unit disk, with a finite sum $\sum_{x \in \mathcal{I}} w_x g(x)$, for a set of points $\mathcal{I} \subset D$ and each with weight $w_x$. We use a simple method of selecting $\mathcal{I}$ and $w_x$. We desire a rotational symmetry, so the points lie on $N_R$ radial rings, each with $N_\Theta$ points.

First, we use Gauss-Legendre quadrature (Hildebrand, 1987) to obtain $N_R$ quadrature points $u$ with weights $w_u$ over the interval $[-1, 1]$. We map this to radial weights and points by $r = \sqrt{(u+1)/2}, w_r = w_u/2$. For the angular component, we take a uniform grid of $N_\Theta$ points on $(-\pi, \pi]$. The integration points on the disk are then $x = (r, \theta)$ with weights $w_x = w_r/N_\Theta$.

This gives accurate integration, because points $(u + 1)/2$ and weights $w_u/2$ create a quadrature scheme on $[0, 1]$. The square root creates a uniform polar grid, as the integration measure for the disk is $r dr d\theta = du d\theta / 2$.

# D SOLVING THE $SO(2)$ KERNEL CONSTRAINT

| $m, n$ | $d(\rho_m \otimes \rho_n)$ | Linearly independent solutions for $K(\theta)$ |
|---|---|---|
| $0, 0$ | $0$ | $1$ |
| $0, n$ | $\begin{pmatrix} 0 & -n \\ n & 0 \end{pmatrix}$ | $(\cos n\theta \quad \sin n\theta), (\sin n\theta \quad -\cos n\theta)$ |
| $m, 0$ | $\begin{pmatrix} 0 & -m \\ m & 0 \end{pmatrix}$ | $\begin{pmatrix} \cos m\theta \\ \sin m\theta \end{pmatrix}, \begin{pmatrix} \sin m\theta \\ -\cos m\theta \end{pmatrix}$ |
| $m, n$ | $\begin{pmatrix} 0 & -n & -m & 0 \\ n & 0 & 0 & -m \\ m & 0 & 0 & -n \\ 0 & m & n & 0 \end{pmatrix}$ | $\begin{pmatrix} c_- & -s_- \\ s_- & c_- \end{pmatrix}, \begin{pmatrix} s_- & c_- \\ -c_- & s_- \end{pmatrix}, \begin{pmatrix} c_+ & s_+ \\ s_+ & -c_+ \end{pmatrix}, \begin{pmatrix} -s_+ & c_+ \\ c_+ & s_+ \end{pmatrix}$ |

Table 7: Solutions to the kernel constraint for kernels that map from $\rho_n$ to $\rho_m$. We denote $c_\pm = \cos(m \pm n)\theta, s_\pm = \sin(m \pm n)\theta$.

Consider a kernel $K$ a function on the tangent space $T_pM$, such that for all $v \in T_pM$, $K(v)$ is a linear map from features in representation $\rho_{\text{out}}$ to features in a representation $\rho_{\text{out}}$. This kernel is equivariant if $\forall v \in T_pM$ and $\forall g \in SO(2)$ the following kernel constraint is satisfied:

$$K(g^{-1}v) = \rho_{\text{out}}(g^{-1})K(v)\rho_{\text{in}}(g). \quad \forall v \in T_pM, g \in SO(2) \tag{8}$$

We immediately see that this constraint is independent for each radius, so we can solve the constraint independently for each radial ring, giving rise to the angular constraint on each ring:

$$K(\theta - g) = \rho_{\text{out}}(-g)K(\theta)\rho_{\text{in}}(g). \quad \forall \theta \in S^1, g \in SO(2) \tag{9}$$

where we note that the circle equals $SO(2)$ and the action of $SO(2)$ on itself is simply an addition of angles. Any representation $\rho$ can be written as equivalent to a direct sum of the irreducible representations (irreps) of $SO(2)$, which are the following, indexed by their order $n \in \mathbb{N}$:

$$\rho_0(g) = 1, \quad \rho_n(g) = \begin{pmatrix} \cos ng & -\sin ng \\ \sin ng & \cos ng \end{pmatrix}, \ n \in \mathbb{N}_{>0}$$

This means that for any representation $\rho$, we can always find an invertible matrix $A$ such: $\forall g : \rho(g) = A^{-1} \text{block\_diag}(\rho_{n_1}, ... \rho_{n_N})A$, where $\{n_1, ..., n_N\}$ indicate the orders of the constituent irreps. Hence, we can solve the angular kernel constraint (Eq. 9) for two irreps, $\rho_{\text{in}} = \rho_n$ and $\rho_{\text{out}} = \rho_m$, and construct the solution for general representations $\rho_{\text{in}}$ and $\rho_{\text{out}}$ from the obtained solutions. Noting that all irreps are orthogonal in our chosen basis are orthogonal, so that $\rho_n(g)^T = \rho(-g)$, we see that, introducing explicit indices:

$$\begin{aligned} K(\theta - g)_{ij} &= \rho_m(-g)_{il}K(\theta)_{lk}\rho_n(g)_{kj} \\ &= \rho_m(-g)_{il}\rho_n(-g)_{jk}K(\theta)_{lk} \\ &= (\rho_m \otimes \rho_n)(-g)_{(ij),(lk)}K(\theta)_{lk} \end{aligned}$$

As this equation is ought to be satisfied for any $g \in SO(2)$ and $SO(2)$ is a one-dimensional Lie group, we can equivalently require it to be satisfied by an infinitesimal element of $SO(2)$ close to the identity, creating an ordinary differential equation:

$$\dot{K}(\theta) = d(\rho_m \otimes \rho_n)K(\theta),$$
$$\text{where} \quad d(\rho_m \otimes \rho_n) := \left( \frac{\partial}{\partial g}(\rho_m \otimes \rho_n)(g) \right)|_{g=0},$$

For various $m, n$, we can construct $(\rho_m \otimes \rho_n)(g)$, take the derivative with respect to $g$ and set $g$ to 0 to obtain the matrix $d(\rho_m \otimes \rho_n)$. Subsequently, any computer algebra system can solve the ODE. The

solutions can be found in Table 7. Now, for a general $\rho_{\text{in}}(g) = A^{-1} \text{block\_diag}(\rho_{n_1}, ...\rho_{n_N})(g)A$, $\rho_{\text{out}}(g) = B^{-1} \text{block\_diag}(\rho_{m_1}, ...\rho_{m_M})(g)B$, a general solution has the form:

$$K(\theta) = B^{-1} \begin{pmatrix} K_{m_1,n_1}(\theta) & \cdots & K_{m_1,n_N}(\theta) \\ \vdots & \ddots & \vdots \\ K_{m_M,n_1}(\theta) & \cdots & K_{m_M,n_N}(\theta) \end{pmatrix} A$$

where $K_{m,n}(\theta)$ is a solution from Table 7 mapping from irrep $\rho_n$ to $\rho_m$. For each in/out irrep pair, we get a linear solution space of 1, 2 or 4 dimensions. For general $\rho_{\text{in}}, \rho_{\text{out}}$, the solution space is the product of the solution spaces of each in/out irrep pair. A parameterized kernel can thus be constructed by assigning a weight to each independent solution of each in/out irrep pair.

# E    EQUIVARIANCE ERROR BOUNDS ON REGULAR NON-LINEARITY

The regular non-linearity acts on each point on the sphere in the following way. For simplicity, we assume that the representation is $U$ copies of $\rho_0 \oplus \rho_1 \oplus ... \oplus \rho_M$. One such copy can be treated as the discrete Fourier modes of a circular signal with band limit $M$. We map these Fourier modes to $N$ spatial samples with an inverse Discrete Fourier Transform (DFT) matrix. Then apply to those samples a point-wise non-linearity, like ReLU, and map back to the Fourier modes with a Discrete Fourier Transform Matrix.

This procedure is exactly equivariant for gauge transformation with angles multiple of $2\pi/N$, but approximately equivariant for small rotations in between.

In equations, we start with Fourier modes $x_0, (x_\alpha(m), x_\beta(m))_{m=1}^B$ at some point on the sphere and result in Fourier modes $z_0, (z_\alpha(m), z_\beta(m))_{m=1}^B$. We let $t = 0, ..., N-1$ index the spatial samples.

$$x(t) = x_0 + \sum_m x_\alpha(m) \cos\left(\frac{2\pi}{N} mt\right) + \sum_m x_\beta(m) \sin\left(\frac{2\pi}{N} mt\right)$$

$$y(t) = f(x(t))$$

$$z_0 = \frac{1}{N} \sum_t y(t)$$

$$z_\alpha(m) = \frac{2}{N} \sum_t \cos\left(\frac{2\pi}{N} mt\right) y(t)$$

$$z_\beta(m) = \frac{2}{N} \sum_t \sin\left(\frac{2\pi}{N} mt\right) y(t)$$

Note that Nyquist's sampling theorem requires us to pick $N \geq 2B + 1$, as otherwise information is always lost. The normalization is chosen so that $z_\alpha(m) = x_\alpha(m)$ if $f$ is the identity.

Now we are interested in the equivariance error between the following two terms, for small rotation $\delta \in [0, 1)$. Any larger rotation can be expressed in a rotation by a multiple of $2\pi/N$, which is exactly equivariant, followed by a smaller rotation. We let $z_\alpha^{FT}(m)$ be the resulting Fourier mode if first the input is gauge-transformed and then the regular non-linearity is applied, and let $z_\alpha^{TF}(m)$ be the result of first applying the regular non-linearity, followed by the gauge transformation.

$$z_\alpha^{FT}(m) = \frac{2}{N} \sum_t \cos\left(\frac{2\pi}{N} mt\right) y(t+\delta) = \frac{2}{N} \sum_t c_m(t) y(t+\delta)$$

$$z_\alpha^{TF}(m) = \frac{2}{N} \sum_t \cos\left(\frac{2\pi}{N} m(t-\delta)\right) y(t) = \frac{2}{N} \sum_t c_m(t-\delta) y(t)$$

where we defined for convenience $c_m(t) = \cos(2\pi mt/N)$. We define norms $||x||_1 = |x_0| + \sum_m (|x_\alpha(m)| + |x_\beta(m)|)$ and $||\partial x||_1 = \sum_m m(|x_\alpha(m)| + |x_\beta(m)|)$.

**Theorem 1.** *If the input $x$ is band limited by $B$, the output $z$ is band limited by $B'$, $N$ samples are used and the non-linearity has Lipschitz constant $L_f$, then the error to the gauge equivariance of the regular non-linearity bounded by:*

$$||z^{FT} - z^{TF}||_1 \leq \frac{4\pi L_f}{N}\left((2B' + \frac{1}{2})||\partial x||_1 + B'(B'+1)||x||_1\right)$$

*which goes to zero as $N \to \infty$.*

*Proof.* First, we note, since the Lipschitz constant of the cosine and sine is 1:

$$|c_m(t-\delta) - c_m(t)| \leq \frac{2\pi m\delta}{N} \leq \frac{2\pi m}{N}$$

$$|x(t+\delta) - x(t)| \leq \frac{2\pi}{N} \sum_m m(|x_\alpha(m)| + |x_\beta(m)|) = \frac{2\pi}{N}||\partial x||_1$$

$$|y(t+\delta) - y(t)| \leq L_f \frac{2\pi}{N}||\partial x||_1$$

$$|c_m(t)| \leq 1$$

$$|x(t)| \leq |x_0| + \sum_m (|x_\alpha(m)| + |x_\beta(m)|) = ||x||_1$$

$$|y(t)| \leq L_f ||x||_1$$

Then:

$$
\begin{aligned}
&|c_m(t)y(t+\delta) - c_m(t-\delta)y(t)| \\
=&|c_m(t)\left[y(t+\delta) - y(t)\right] - y(t)\left[c_m(t-\delta) - c_m(t)\right]| \\
\leq&|c_m(t)||y(t+\delta) - y(t)| + |y(t)||c_m(t-\delta) - c_m(t)| \\
\leq& L_f \frac{2\pi}{N}||\partial x||_1 + L_f ||x||_1 \frac{2\pi m}{N} \\
=& \frac{2\pi L_f}{N}\left(||\partial x||_1 + m||x||_1\right)
\end{aligned}
$$

So that finally:

$$
\begin{aligned}
&|z_\alpha^{FT}(m) - z_\alpha^{TF}(m)| \\
\leq& \frac{2}{N}\sum_t |c_m(t)y(t+\delta) - c_m(t-\delta)y(t)| \\
\leq& \frac{4\pi L_f}{N}\left(||\partial x||_1 + m||x||_1\right)
\end{aligned}
$$

The sinus component $|z_\beta^{FT}(m) - z_\beta^{TF}(m)|$ has the same bound, while $|z_0^{FT} - z_0^{TF}| = |y(t+\delta) - y(t)|$, which is derived above. So if $z$ is band-limited by $B'$:

$$
\begin{aligned}
||z^{FT} - z^{TF}||_1 &= |z_0^{FT} - z_0^{TF}| + \sum_{m=1}^{B'} |z_\alpha^{FT}(m) - z_\alpha^{TF}(m)| + |z_\beta^{FT}(m) - z_\beta^{TF}(m)| \\
&\leq \frac{4\pi L_f}{N}\left((2B' + \frac{1}{2})||\partial x||_1 + \sum_{m=1}^{B'} 2m||x||_1\right) \\
&= \frac{4\pi L_f}{N}\left((2B' + \frac{1}{2})||\partial x||_1 + B'(B'+1)||x||_1\right)
\end{aligned}
$$

Since $||\partial x||_1 = \mathcal{O}(B||x||_1)$, we get $||z^{FT} - z^{TF}||_1 = \mathcal{O}(\frac{BB' + B'^2}{N}||x||_1)$, which obviously vanishes as $N \to \infty$. $\qquad \square$

# F  EXPERIMENTAL DETAILS

In the following subsections, we provide with more details regarding experiments on MNIST, Atomization Energy Prediction and Cosmological Model Classification.

## F.1  MNIST

For the MNIST experiments, we disambiguate two variants, both used with interpolation and without. The first variant uses only the 1-ring of 7 nearest neighbors and representations $\rho_0 \oplus \rho_1$. This version uses three radial functions: one that covers the self-interaction and one that covers the other neighbors.

The second variant uses the 2-ring of 19 nearest neighbors and representations $\rho_0 \oplus \rho_1 \oplus \rho_2$. This version uses three radial functions: one for the self-interaction, one for the neighbors on the 1-ring and one for the neighbors on the 2-ring.

For either variant, we build a network consisting of 8 convolutional layers, with stride after each 2 layers. A stride layer selects the vertices that are present in the IcoSphere with one fewer subdivision, and divides the number of vertices by roughly 4. The input is a scalar feature. After convolution, the irrep features with multiplicity after each layer: [16, 16, 32, 32, 64, 64, 64, 64]. The last convolution layer maps to a scalar feature of multiplicity 64. Then the signal is averaged over the vertices, creating a gauge invariant network. This is then processed by 2 layer of MLP mapping to multiplicity 50 and then to the 10 classification logits.

Nonlinearities are ReLU for scalar features and the `RegularNonlinearity` for all other features. Batchnorm is used, as well as dropout with drop probability 10%.

## F.2  ATOMIZATION ENERGY PREDICTION

Our network architecture used in this experiment is depicted in Table 8. We use four blocks of gauge equivariant convolutions followed by batchnorm and regular nonlinearities. We also deploy pooling / downsampling layers in blocks (1-3) to reduce the resolution of the spherical maps progressively. Rest of the network consists of linear layers followed by batchnorm and ReLU. We sum-pool over the atoms before the loss function (see block-7) to become invariant against permutations of them.

We train the model with batch size 32 and an initial learning rate of $1\mathrm{e}{-3}$ for 30 epochs using mean-squared-error (MSE) as loss function. The learning rate is reduced by $0.1$ every tenth epoch.

| block | Operators | Multiplicity (in/out) | Order ($\rho_{in}/\rho_{out}$) |
|---|---|---|---|
| 1 | [Convolution, Downsample, BatchNorm, Regular Nonlinearity] | 5/16 | 0/1 |
| 2 | [Convolution, Downsample, BatchNorm, Regular Nonlinearity] | 16/32 | 1/2 |
| 3 | [Convolution, Downsample, BatchNorm, Regular Nonlinearity] | 32/64 | 2/1 |
| 4 | [Convolution, BatchNorm, ReLU] | 64/64 | 1/0 |
| 5 | [Linear, BatchNorm1d, ReLU] | 64/256 | 0/0 |
| 6 | [Linear, BatchNorm1d, ReLU] | 256/64 | 0/0 |
| 7 | AtomPool | 64/64 | 0/0 |
| 8 | [Linear, BatchNorm1d, ReLU] | 64/512 | 0/0 |
| 9 | [Linear, BatchNorm1d, ReLU] | 512/1 | 0/0 |

Table 8: Network architecture used in Atomization Energy Prediction experiment.

## F.3  COSMOLOGICAL MODEL CLASSIFICATION

For this experiment, we use HEALpix sampling on split level $o = 4$ following Perraudin et al. (2018) which covers $0.5\%$ of the spherical manifold and has signal dimensionality of 65k.

For the experiments, we use a network architecture identical to the one described in (Perraudin et al., 2018) by replacing their graph convolutions with our gauge equivariant convolutions. We elaborate the architecture in Table 9.

For each noise factor, we train an instance of this model with batch size 16 and an initial learning rate of $1e-4$ for 120 epochs. Finally, we use Binary Cross Entropy as the loss function.

| block | Operators | Multiplicity (in/out) | Order ($\rho_{in}/\rho_{out}$) |
|---|---|---|---|
| 1 | [Convolution, Downsample, BatchNorm, Regular Nonlinearity] | 1/10 | 0/1 |
| 2 | [Convolution, Downsample, BatchNorm, Regular Nonlinearity] | 10/20 | 1/1 |
| 3 | [Convolution, Downsample, BatchNorm, Regular Nonlinearity] | 20/40 | 1/1 |
| 4 | [Convolution, Downsample, BatchNorm, Regular Nonlinearity] | 40/40 | 1/1 |
| 5 | [Convolution, Downsample, BatchNorm, Regular Nonlinearity] | 40/40 | 1/1 |
| 6 | [Convolution, Downsample, BatchNorm] | 40/2 | 1/0 |
| 7 | Softmax | 2/2 | 0/0 |

Table 9: Network architecture used in Cosmological Model Classification experiment.

