# OpenReview forum: "Gauge Equivariant Spherical CNNs"
_ICLR.cc/2020/Conference — Reject_

### Official Review · AnonReviewer1 · 2019-10-23
**Official Blind Review #1**

**Rating:** 8

**Review:**

The paper proposes SO(3) equivariant layer, derived from the recently introduced Gauge equivariant CNN framework. The novel contributions are in taking the Gauge equivariance CNN and finding efficient ways to perform logarithmic mapping, parallel transport, and convolution by the equivariant kernel when applied to the sphere. An interpolation scheme for an improved approximation to global SO(3) symmetry is also discussed. Experimental results on spherical MNIST, climate pattern segmentation, and omnidirectional semantic segmentation demonstrate the usefulness of the proposed method for prediction on a sphere.

For the most part, the paper is very clearly written despite the challenging and technical nature of the topic. In particular, the first four pages provide a nice overview of related work and a clear explanation of the Gauge equivariance framework of Cohen et al’19. Two sections that can benefit from further clarification are the proposed "regular non-linearities" (where it would be nice to show in equation why we have equivariance), and equation (4), the logarithmic map (where I had a hard time mapping the discussion in words to the equation).

However, the experiments, while satisfactory, are not impressive: one issue is that the results are mostly compared to the results of two relevant papers by Cohen and colleagues. In recent years, there have been other proposals for deep learning on the sphere, and I wonder why experiments do not try to compare with these works?  (see Kondor et al’18, Coors et al’18, and others cited in the paper.) Moreover, although in theory, the proposed framework improves the Icosahedral CNN of Cohen et al’19 (by directly operating on the sphere rather than an Icosahedral approximation), the practical improvements over the Icosahedral CNN seem to be often marginal (with one exception in spherical MNIST). Do you have any explanation for this? Is there any setup where you expect the proposed approach would give a substantial improvement?


**Experience Assessment:**

I have published in this field for several years.

**Review Assessment: Checking Correctness Of Derivations And Theory:**

I assessed the sensibility of the derivations and theory.

**Review Assessment: Checking Correctness Of Experiments:**

I assessed the sensibility of the experiments.

**Review Assessment: Thoroughness In Paper Reading:**

I read the paper at least twice and used my best judgement in assessing the paper.

---

> ### Author Response · Authors · 2019-11-13
> **Response to reviewer #1**
>
> Thank you for your comments and kind words on the readability of our paper.
>
> - We agree that the regular non-linearities and log map could be clarified more and attempted to do so in the revised version. We have added a proof that shows in the limit of infinite samples in the regular non-linearity, we achieve full equivariance and provide bounds on the equivariance error for finite number of samples.
> - We have added a comparison to Kondor et al’18 to our MNIST experiment and further comparisons in the additional experiments.
> - We agree that the difference between the IcoCNN is marginal, except crucially for a task that directly tests equivariance, which is the NR/R experiment for rotated MNIST. This experiment shows that the icosahedral approximation leads to poor equivariance under SO(3) rotations. The other experiments don’t test for generalisation to arbitrary rotations and thus show less difference between the IcoCNN and our method.

---

### Official Review · AnonReviewer2 · 2019-10-24
**Official Blind Review #2**

**Rating:** 8

**Review:**

1. (p1) ``in almost in all cases" to "in almost all cases"
2.  (1.1) The authors could explain more about why we would want to consider tensor features.
3. They conducted experiments on different datasets, including the MNIST dataset. They achieved good results comparing to baseline spherical CNNs. However, the advantage of this method over S2CNN can be further elaborated, as S2CNN already achieved high accuracy; it seems like the one improvement is the complexity (improved from S2CNN's $O(N \log N)$ to their model's $O(N)$), but the reduction of complexity is not significantly reflected in the training time per epoch (from 380s to 284 s).
4. Overall, the paper provides clear theoretical backgrounds on gauge CNNs that justifies their definition of convolution operator only uses the intrinsic structure of the manifold (does not reply on higher dimensional embedding).

**Experience Assessment:**

I have published one or two papers in this area.

**Review Assessment: Checking Correctness Of Derivations And Theory:**

I assessed the sensibility of the derivations and theory.

**Review Assessment: Checking Correctness Of Experiments:**

I assessed the sensibility of the experiments.

**Review Assessment: Thoroughness In Paper Reading:**

I read the paper at least twice and used my best judgement in assessing the paper.

---

> ### Author Response · Authors · 2019-11-13
> **Response to reviewer #2**
>
> Thank you for your comments.
>
> 1. We have fixed this typo, thank you for pointing it out.
> 2. When an equivariant method uses only scalar features, it necessarily must use isotropic kernels. This can be seen from Case III in Proposition 1 of [1]. This limits expressivity and complicates the detection of orientable features, such as lines. We show this empirically with the Isotropic baseline in the MNIST experiment. The main motivation for using non-scalar features is that it allows for equivariant anisotropic kernels. Additionally, some datasets consist of non-scalar signals, like optical flows, SIFT-like features that measure local properties in different directions, wave polarization, or wind direction. In these cases one has no choice but to use non-scalar features in the input space.
> 3. The higher computational complexity of S2CNN [2] prevents scaling up to high resolution grids, as noted by [3]. Our method does not suffer from this limitation. The fact that in the MNIST experiments, which use a relatively coarse grid, the runtimes are quite similar, is possibly due to the custom CUDA kernels used in [2]. We expect custom kernels can yield big improvements in the runtime of our method and are investigating possible implementations.
>
> [1] Kondor & Trivedi 2018
> [2] Cohen et al. 2018
> [2] Perraudin et al. 2019

---

### Official Review · AnonReviewer3 · 2019-10-25
**Official Blind Review #3**

**Rating:** 3

**Review:**

Cohen et al. recently proposed the "Gauge equivariant CNN" framework for generalizing convolutions to arbitrary differentiable manifolds. The present paper instantiates this framework for the case of the sphere.

The sphere is the simplest natural non-trivial manifold to try out gauge invariant networks on, and spherical CNNs have several applications. However, other than the details of the interpolation etc., there is really very little in this paper that is new relative to the original paper by Cohen et al., it reads a bit more like an extended "experiments" section.

Unfortunately the experimental results are not all that remarkable either, probably because the tasks are relatively easy, so other SO(3) equivariant architectures do quite well too. Given that there is essentially no new theory in the paper, I would have welcomed a much more thorough experimental section, comparing different architectures, different discretization strategies of the sphere and different interpolations/basis functions.

**Experience Assessment:**

I have published in this field for several years.

**Review Assessment: Checking Correctness Of Derivations And Theory:**

I assessed the sensibility of the derivations and theory.

**Review Assessment: Checking Correctness Of Experiments:**

I assessed the sensibility of the experiments.

**Review Assessment: Thoroughness In Paper Reading:**

I read the paper thoroughly.

---

> ### Author Response · Authors · 2019-11-13
> **Response to reviewer #3**
>
> Thank you for your comments. It is true that the present paper does not introduce a new framework like the paper by Cohen et al. However, although that paper contained a detailed continuous mathematical theory, as well as a discretized implementation of the idea for the icosahedron, it lacked an explanation of how exactly the method is to be implemented for manifolds that are not locally flat like the icosahedron. This is certainly not a trivial matter, as there are many ways one might derive a discrete algorithm from the continuous theory, all of which will have different characteristics in terms of runtime efficiency, numerical accuracy, and task performance. After careful consideration and preliminary experiments, we have settled on the convolution algorithm described in the paper, which is based on a non-trivial interpolation scheme.
>
> Nevertheless, the reviewer is correct that this is not a theoretical paper. In order to strengthen the theoretical side of the paper, we have added appendix E, which contains a theoretical analysis of the equivariance of the regular non-linearity under SO(2) gauge transformations. It shows that pointwise nonlinearities, which tend to perform best, can be used for continuous groups and the number of samples can be selected to trade off computational cost with the equivariance error.
>
> We have also worked to further strengthen the experimental section of the paper. We added comparisons to more prior work and an isotropic baseline to the Spherical MNIST; we demonstrate state of the art on molecular energy prediction; and demonstrate scalability on a high resolutions cosmology dataset - a task Fourier based spherical convolutions would fail at. We hope that the additional experiments added to the revised paper, such as the baseline using only scalar features and isotropic filters, help showing the strengths and weaknesses of various convolutional methods on the sphere.

---

### Author Response · Authors · 2019-11-13
**Overview of revisions and responses**

We thank the reviewers for their valuable comments, which we have taken into account in our revised version.

First, we would like to stress the difference between our method and other equivariant spherical convolution methods:
- Isotropy vs Anisotropy: Methods using only scalar features are restricted to using isotropic filters in order to be SO(3) equivariant, which includes graph methods such as [1]. Alternatively, one can pool over orientations immediately after convolution (as in [2]). In either case, however, detecting the direction of orientable patterns such as lines is complicated. We show this empirically in the isotropic baseline we added to our revised version (see below).
- Computational Efficiency: Fourier based methods, such as [3] and [5], are automatically SO(3) equivariant, but scale poorly to high resolution grid, due to the nonlinear complexity of the Fourier transform and various difficulties implementing it efficiently in current deep learning frameworks. Additionally, they can’t be easily applied to grids on part of the sphere.
- Icosahedral CNN: The Icosahedral CNN [4] is fast to compute thanks to the use of conv2d routines and exactly gauge equivariant and as a result automatically equivariant up to 60 discrete symmetries of the icosahedron. However, it is not equivariant to SO(3), while our model is fully SO(3)-equivariant. Additionally, Icosahedral CNNs assume a particular sampling grid whereas our Spherical CNN can admit arbitrary grids. This feature is particularly important when considering the fact that in many applications spheres are discretized differently than the one Icosahedral CNNs and others presume.  Finally, it is not straightforward to adjust Icosahedral CNNs to operate over partially observed spherical inputs in an efficient manner as our method.

All in all, to the best of our knowledge, our method is the first spherical convolution which is SO(3) equivariant, supports anisotropic filters, and scales to arbitrary high resolution grids.

Secondly, the reviewers have pointed out that the experimental validation could be improved. To address this, we add two new experiments to the revised version. With these experiments, we would like to emphasize both scalability and flexibility aspects of our method using a different sampling strategy and larger dimensional spherical signals as well as cases where our anisotropic filters could potentially lead to better performance in comparison to isotropic counterparts. We list our experimental revisions below.

- Additional Experiment #1 (Atomization Energy Prediction): In this dataset, we achieve state of the art compared to other sphere-based methods, in order to expand the experimental validation of our method.
- Additional Experiment #2 (Cosmological Model Classification): The resolution of the signals in this problem are a few orders of magnitude larger in comparison to the existing spherical datasets.  Also, it requires a different spherical sampling scheme namely, Healpix. Thus we use it to demonstrate our approach's scalability and grid-agnostic aspects.

In addition to those, the following aspects have changed in the revised version:
- We added a comparison to [5] in our Spherical MNIST experiment, as requested by reviewer #1.
- We added a baseline to our Spherical MNIST experiment in which we use only scalar features and isotropic filters. The results show that anisotropic filters are important in this task, as mentioned by reviewer #2.
- We prove a bound on the error to the gauge equivariance of the Regular NonLinearity in Appendix E, as requested by reviewer #1.
- We clarified the log map, as requested by reviewer #1. We moved the sections regarding the spherical geometry to the appendix to make space for the additional experiments.
- We added a figure of the icosphere with exponential/log map to visually support equations as requested by reviewer #1.

[1] Perraudin et al. 2019
[2] Masci et al. 2015
[3] Cohen et al. 2018
[4] Cohen et al. 2019
[5] Kondor et al. 2018

---

### Decision · Program_Chairs · 2019-12-19

**Decision:**

Reject

**Comment:**

The paper extends Gauge invariant CNNs to Gauge invariant spherical CNNs.  The authors significantly improved both theory and experiments during the rebuttal and the paper is well presented. However, the topic is somewhat niche, and the bar for ICLR this year was very high, so unfortunately this paper did not make it. We encourage the authors to resubmit the work including the new results obtained during the rebuttal period.